# Thermal-Hydraulic Analysis of Parabolic Trough Collectors Using Straight Conical Strip Inserts with Nanofluids

**DOI:** 10.3390/nano11040853

**Published:** 2021-03-26

**Authors:** Nabeel Abed, Imran Afgan, Hector Iacovides, Andrea Cioncolini, Ilyas Khurshid, Adel Nasser

**Affiliations:** 1Department of Mechanical, Aerospace and Civil Engineering, School of Engineering, University of Manchester, Manchester M13 9PL, UK; h.Iacovides@manchester.ac.uk (H.I.); Andrea.cioncolini@manchester.ac.uk (A.C.); a.g.nasser@manchester.ac.uk (A.N.); 2Mechanical Technical Department, Technical Institute of Anbar, Middle Technical University, Baghdad 10066, Iraq; 3Department of Mechanical Engineering, College of Engineering, Khalifa University, Abu Dhabi 12277, United Arab Emirates; ilya.khurshid@ku.ac.ae

**Keywords:** heat transfer, swirl generators, non-uniform heating, parabolic solar trough collectors, solar thermal energy, thermal and hydraulic performance

## Abstract

In this study, we numerically investigated the effect of swirl inserts with and without nanofluids over a range of Reynolds numbers for parabolic trough collectors with non-uniform heating. Three approaches were utilized to enhance the thermal-hydraulic performance—the variation of geometrical properties of a single canonical insert to find the optimized shape; the use of nanofluids and analysis of the effect of both the aforementioned approaches; the use of swirl generators and nanofluids together. Results revealed that using the straight conical strips alone enhanced the Nusselt number by 47.13%. However, the use of nanofluids along with the swirl generators increased the Nusselt number by 57.48%. These improvements reduced the thermal losses by 22.3% for swirl generators with nanofluids, as opposed to a reduction of only 15.7% with nanofluids alone. The investigation of different swirl generator designs showed various levels of improvements in terms of the overall thermal efficiency and thermal exergy efficiency. The larger swirl generator (H30mm-θ30°-N4) with 6% SiO_2_ nanofluids was found to be the optimum configuration, which improved the overall collector efficiency and thermal exergy by 14.62% and 14.47%, respectively.

## 1. Introduction

The use of non-renewable fossil fuels (coal, oil and gas) has led to the social-economic growth of the whole world. However, the consumption of these fossil fuels has a negative impact on environment, climate and human health. Therefore, it is imperative to move to renewable sources of energy that are continually replenished by nature (hydro, wind, solar, geothermal, wave, tidal, biomass, etc.). Among these renewable energy sources, solar energy is widely available as it directly taps the immense power of the sun to produce heat and electricity. Solar energy can be harnessed by concentrating solar power (CSP) plants. In CSP plants, mirrors are used to concentrate the sunlight onto a receiver that collects and transfers the solar energy to a heat-transfer fluid. The variants of CSP include linear Fresnel reflectors, parabolic trough collectors (PTCs), solar towers and parabolic dishes [1]. Among the above-mentioned types of CSP, the most efficient, cost effective and commercially developed is the PTC because of its capability to work at moderate temperatures, which are suitable for industrial and engineering applications.

The thermal and hydraulic performance of PTC systems can be improved by investigating the effect of various pure working fluids with different operating conditions [2,3,4,5]. Other practices used to increase PTC performance are mixing working fluids and nanoparticles with thermal properties that are better than the base-pure working fluids. This technique could increase the Prandtl number (the ratio of momentum diffusivity to thermal diffusivity). Fluids with larger Prandtl numbers are free flowing, with higher thermal conductivity than the pure working fluids, and are thus a better choice for enhancing the thermal energy performance [6]. The use of nanofluids is a promising technique to improve the thermal conductivity, as it enhances the Nusselt number (the ratio of convective to conductive heat transfer across a boundary), thermal efficiency and exergy efficiency. More information about nanofluids and its application can be found in [7,8,9,10,11,12,13].

The insertion of different types of metallic turbulators inside the solar absorber in the flow path can increase the absorption of thermal energy from the internal absorber wall. This process enhances the thermal conductivity of turbulators and the surface area for heat transfer. Furthermore, this technique decreases the external absorber wall temperature, resulting in lower thermal losses and higher thermal efficiencies. According to Reddy et al. [14], using two different types of fins (longitudinal fins and porous fins) could increase the heat transfer by 17.5% compared to a pure absorber. Ravi and Reddy [15] numerically investigated the effect of the porous disc with different configurations (bottom half disc, full porous disc, bottom half porous disc and typical receiver). They found that the Nusselt number was enhanced by 64.3% by using the receiver with the top porous disc at the expense of an increased pressure drop of 457 Pa.

After placing the milt-longitudinal vortexes on the inner side of the receiver of a PTC system, Cheng et al. [16] found that thermal losses and wall temperature initially went down. However, when the Reynolds number was increased, the thermal losses were reduced by 1.35–12.1%. Thus, the augmentation in the performance evaluation criteria was observed to be 1.18%. Additionally, Wang et al. [17] observed that when they used metal foam as an insert type inside the solar absorber, the Nusselt number, friction factor and performance evaluation criteria increased by 10–12 times, 400–700 times and 1.1–1.5 times, respectively, compared to a standard tubular receiver. Another type of insert—perforated plate inserts—was investigated by Mwesigye et al. [18]. They observed that the Nusselt number, thermal enhancement factor and friction factor all improved substantially compared to an empty receiver. Other notable contributions using different types of inserts are summarised in Table 1.

All cited previous studies investigated various types of inserts under different operating conditions and non-uniform heat flux distributions around the solar absorber on the external surface. However, optimized straight conical strip inserts have never been investigated thoroughly. Thus, in this study, the effects of optimized straight conical strip inserts attached to the central rod are investigated under a non-uniform heat flux profile around the absorber tube. We assumed the central rod diameter to be constant (D_r_ = 1.6 cm) and varied the height of the attached strip (H) from 15 mm to 30 mm with the angle of the attached strip (θ) changed from 15° to 30°. All arrangements were then examined for a range of Reynolds numbers (10^4^ to 10^5^). After obtaining the optimum strip height and angle, the number of strips were increased stepwise from one to four. The results were then examined with the presence of a nanofluid using 6% of SiO_2_ nanoparticles mixed in Therminol^®^ VP-1, as this was the best candidate proposed by Abed et al. [13]. It is important to mention that all case studies were assumed to be steady-state, incompressible and three dimensional, using Therminol^®^ VP-1 as a heat transfer fluid with an inlet temperature of 400 K. The heat flux distribution was taken as a realistic profile by applying the output of the Monte Carlo ray tracing (MCRT) model in the circumferential direction. The study evaluates and assesses the influence of the optimum case on the thermal, hydraulic and thermodynamic performances of the PTCs. To the authors’ knowledge, such an evaluation of PTC performance has not yet been reported in the literature.

## 2. Background and Mathematical Model of PTCs

A parabolic trough collector (PTC) is made up of long, linear parabolic-shaped concentrating system of mirrors and a receiver tube that is placed along the focal axis of the parabola. A schematic diagram of the parabolic trough and its receiver is illustrated in Figure 1. It can be observed from the figure that the PTC arrangement consists of a parabolic trough, which is made from mirror shaped like a parabola. The parabolic trough receives the solar energy from the sun and reflects it to the envelope, which is made from a glass. Then, the absorbed thermal energy is transferred by conduction to the absorber tube (also known as the heat collection element (HCE)). The HCE is located in the focal line of the parabola inside the glass envelope. The received heat is transferred by conduction from the outer surface of the absorber tube to the inner surface. Inside the HCE, the heat transfer fluid receives a part of the thermal energy by convection.

The remaining energy is sent back to the outer surface of the HCE, which in turn sends the energy to the internal surface of the glass envelope via convection and radiation. The energy continues transferring from the internal surface of the glass envelope (a) to the external surface of the glass envelope via conduction (b) from the external surface to the ambient via convection and (c) to the surrounding via radiation as losses, as shown in the thermal resistance model presented in Figure 2.

### 2.1. Geometrical, Hydraulic and Thermal Mathematical Expressions

The geometrical profile of the PTC is represented by the following parabola expression based on the x and y Cartesian co-ordinates, from Duffie and Beckman [36]:
(1)$$x^{2}=4yf_{L}$$
where $f_{L}$ is the focal line, which can be calculated from the following formula based on the aperture width $w_{a}$ of the trough and the rim angle $\varphi_{r}$, from [37]:(2)$$f_{L}=\frac{w_{a}}{4\;\tan\left(\frac{\varphi_{r}}{2}\right)}$$

The thermal efficiency $\eta_{th}$ of the PTC system can be determined based on the useful gained thermal energy $Q_{u}$ and available solar energy $Q_{s}$ as follows:(3)$$\eta_{th}=\frac{Q_{u}}{Q_{s}}$$

The useful gained thermal energy $Q_{u}$ is determined from the following formula based on the mass flow rate of the HTF $\dot{m}$ and fluid temperatures at inlet and outlet $T_{in}$ and $T_{out}$, as per [37]:(4)$$Q_{u}=\dot{m}\;C_{p}\left(T_{out}-T_{in}\right).\;$$

The parameter $C_{p}$ represents the specific heat capacity of the heat transfer fluid at the average temperature ($0.5\;(T_{in\;}$ +$\;T_{out}$)). However, the available solar energy $Q_{s}$ is calculated based on the solar beam radiation $G_{b}$ and the collector aperture area $A_{a}$ as:(5)$$Q_{s}=A_{a}\;G_{b}$$

However, the overall efficiency $\eta_{overall}$ of the PTC system can be determined by taking the pumping power effect into account, as proposed by Wirz et al. [38]:(6)$$\eta_{overall}=\frac{Q_{u}-W_{p}/\eta_{el}}{Q_{s}}$$

The variable $\eta_{el}$ is the electrical efficiency of the power block, which is taken as a typical value, at 0.327. The parameter $W_{p}$ is the pumping power, which can be calculated from the following equation based on the volumetric flow rate $\dot{V}$ and the pressure drop ΔP of the fluid flowing through the solar absorber: (7)$$W_{p}=\Delta P\dot{V}.\;$$

The pressure drop in the solar absorber can be determined based on the friction factor *f* as follows, according to [39]:(8)$$\Delta P=f\frac{L}{D_{i}}\frac{\rho \;U^{2}}{2\;}$$
(9)$$f=\frac{8\tau }{\rho U^{2}\;}$$
where ρ is the fluid density, U the fluid average velocity, $D_{i}$ is the absorber internal diameter, L is the solar tube length and τ is the wall shear stress. One of the most important approaches in assessing and evaluating the thermal performance and frictional effects of any passive technique such as nanofluids, tabulators, etc., is using the performance evaluation criterion (PEC), according to Ralph [40]. The PEC  parameter represents the enhancement in heat transfer and friction factor (or pressure drop) achieved by the passive technique compared with the heat transfer ($Nu_{o}$) and friction factor ($f_{o}$) produced by the plain case based on the same mass flow rate and/or Reynolds number. This parameter can be formulated as:(10)$$PEC=\frac{\left(\frac{Nu}{Nu_{o}}\right)}{{\left(\frac{f}{f_{o}}\right)}^{1/3}}$$

The PEC with a value of more than 1 means that there is some augmentation in heat transfer and friction factor. In terms of thermodynamics analysis, the PTC’s useful output exergy *E_u_* is determined based on the gained useful thermal energy, average heat transfer fluid temperature *T_ave_* and the ambient temperature $T_{am}$ as per Yazdanpanahi et al. [41]:(11)$$E_{u}=\;Q_{u}-m\times C_{p}\;T_{am}\ln\left(\frac{T_{out}}{T_{in}}\right)-m\times T_{am}\frac{\Delta P}{\rho T_{ave}}$$

The third term on the right-hand side of the above relationship can be ignored as the ratios of the pressure drop to density and average temperature are very small. The available solar exergy is then determined from the following model by Petela [42]:(12)$$E_{s}=Q_{s}\;\left[1-\frac{4}{3}\left(\frac{T_{am}}{T_{sun}}\right)+\frac{1}{3}{\left(\frac{T_{am}}{T_{sun}}\right)}^{4}\right].$$
where $T_{sun}$ is the temperature of the external layer of the sun, which is known to be at 5800 K. For the PTC system, the exergy efficiency is defined from the following expression, as per [37]:(13)$$\eta_{ex}=\frac{E_{u}}{E_{s}}$$

In terms of thermal losses $Q_{loss}$ in the PTC system, both convection and radiation heat transfer models take place and the transfer of energy from the outer surface of the glass envelope to the ambient space and its surroundings is calculated by Bhowmik and Mullick [43] as:(14)$$Q_{loss}=L\pi D_{oe}h_{out}\left(T_{oe}-T_{am}\right)+L\pi D_{oe}\;\sigma \varepsilon_{oe}\left(T_{oe}^{4}-T_{sky}^{4}\right)$$
where the subscript oe refers to the outer surface of the glass envelope, σ is the Stefan–Boltzmann constant (5.67 × 10^−8^ W/m^2^ K^4^) and
$\varepsilon_{oe}$ is the glass envelope emissivity on the outer surface. Here, the convection heat transfer coefficient of the ambient $h_{out}$ can be calculated using the following equation based on the wind speed $V_{w}\;$ and the outer glass envelope diameter $D_{oe}$ as suggested by Bhowmik and Mullick [43]:(15)$$h_{out}=4V_{w}^{0.58}D_{oe}^{-0.42}$$

Finally, the sky temperature $T_{sky}$ can be determined based on the ambient temperature using the following formula from Swinbank [44]:(16)$$T_{sky}=\;0.0552\;T_{am}^{1.5}$$

It is important to note here that if the glass envelope is entirely removed from the PTC system (as in the current study), the thermal losses will be directly transferred from the external surface of the heat collection element by convection to the ambient area and by radiation to the surroundings. Therefore, Equation (14) can be replaced by the following expression:(17)$$Q_{loss}=L\pi \;D_{o}h_{out}\left(T_{o}-T_{am}\right)+L\;\pi \;D_{o}\sigma \;\varepsilon_{o}\left(T_{o}^{4}-T_{sky}^{4}\right)$$

Here the subscript o refers to the outer surface of the solar absorber and $\varepsilon_{o}$ is the absorber emissivity on the outer surface, which depends on the average outer surface temperature *T_o_* ($T_{o}$ is in °C) and the selective coatings. For the current model, the expression proposed by Dudley et al. [45] is used as follows.
(18)$$\varepsilon_{o}=0.062+2\times 10^{-7}\times T_{O}^{2}.\;$$

The ambient convective heat transfer coefficient in the case of the bare PTC system reads as
(19)$$h_{out}=4V_{w}^{0.58}D_{o}^{-0.42}$$

The convection heat transfer coefficient h inside the absorber tube is calculated based on the useful thermal energy and the inner wall average temperature $T_{w}$ (which is taken from the simulations) as: (20)$$h=\frac{Q_{u}}{\pi \cdot D_{i}\cdot L\cdot (T_{w}-T_{ave})}$$

The flow is known as fully turbulent in the solar absorber when Re ≥ 4000. The corresponding Nusselt Nu number can then be measured using the following empirical correlation as proposed by Gnielinski [46]:(21)$$Nu=\frac{\left(\frac{f}{8}\right)\left(Re-1000\right)\;Pr}{1+12.7{\left(\frac{f}{8}\right)}^{0.5}\left(Pr^{\frac{2}{3}}-1\right)}for\;\left\{\begin{array}{c} 0.5\leq Pr\leq 2000\\ 3\times 10^{3}<Re<5\times 10^{6} \end{array}\right\}$$

The above equation is used to validate the predicted Nu number from the simulations, where the Reynolds number (Re), Prandtl number (Pr) and Nusselt number (Nu) are defined as:(22)$$Re=\frac{\rho \;U\;D_{i}}{\mu }.\;$$
(23)$$Pr=\frac{\mu \;C_{p}}{k}$$
(24)$$Nu=\frac{hD_{i}}{k}$$

The parameters k and μ are the fluid thermal conductivity and dynamic viscosity, respectively, whereas the tube diameter is taken as the hydraulic diameter in the presence of central rod cases, which can be calculated as ($D\equiv D_{i}-D_{r}$). The friction factor f, predicted from the simulations, is also validated with the following empirical correlation proposed by Petukhov [47] as:(25)$$f={\left(0.75\;ln\;Re^{-1.64}\right)}^{-2}for\;\left\{3000<Re<5\times 10^{6}\right\}$$

### 2.2. Absorber Material and Thermal Fluid Properties

The glass envelope is completely removed in the present work for simplicity. Removing the glass envelop does indeed make the model simpler but glass, while transparent to the incoming solar radiation, is opaque to the outgoing radiation from the collector. This has a strong effect on the thermal equilibrium, also known as the greenhouse effect. As a result, the numerical results will somewhat under-estimate the heat absorbed by the collector, as per [37]. However, since the objective of this study is not to look for absolute values but rather to assess the relative performance of turbulator effects, the removal of the glass envelope satisfies the current modeling requirements. Forristall [3] suggested that using 321H stainless steel material is an appropriate choice for the solar absorber, owing to its strength and resistance to buckling. The geometrical characteristics and environmental conditions, along with the material properties, are presented in Table 2.

The heat transfer fluid used in the current study is called Therminol^®^ VP-1, with an inlet temperature of 400 K. The thermal properties of this working fluid are given in Table 3.

In the experimental work, the preparation of nanofluids can be performed using different methods. In general, there are two main methods that have been proposed in the literature—the one-step approach and the two-step approach. In the first approach, the formations of both nanoparticles and the base fluid are combined together. However, if the nanoparticles are already prepared and are then mixed with the base fluids, the procedure is known as the two-step approach. For numerical simulations, the application of nanofluids can either be a single-phase process or a two-phase process, using the different models proposed in the literature, which depend on the material/fluid properties, such as density, specific heat capacity, thermal conductivity and dynamic viscosity, Michaelides [48]. In order to apply nanofluids numerically, the first step is the nanofluid evaluation, that is, to determine the thermal properties of the nanofluid using the single-phase method which is utilized in the present study. In the literature, several models have been proposed to calculate the thermal properties of nanofluids. Pak and Cho [49] proposed a nanofluid density (*ρ**_nf_*) model based on a heterogeneous mixture, whereas the specific heat capacity ($C_{p,nf}$) model was derived based on the thermal equilibrium between the base fluid and solid particles as proposed by Xuan and Wilfried [50]. The nanofluid dynamic viscosity ($\mu_{nf}$) model proposed by Maiga et al. [51] is used in the current work, which is based on experimental measurements. Moreover, the nanofluid thermal conductivity ($k_{nf}$) model proposed by Bruggeman [52], which is based on a spherical solid–fluid mixture, is applied in the present work. These models are presented below:(26)$$\rho_{nf}=\rho_{s}\varphi +\rho_{f}\left(1-\varphi \right)$$
(27)$$C_{p,nf}=\frac{1}{\rho_{nf}}\left[\rho_{s}C_{p,s}\varphi +C_{p,f}\rho_{f}\left(1-\varphi \right)\right]$$
(28)$$\mu_{nf}=\mu_{f}\left(1+7.3\varphi +123\varphi^{2}\right)$$
(29)$$k_{nf}=0.25\left[\left(3\varphi -1\right)k_{s}+\left(2-3\varphi \right)k_{f}+\sqrt{\Delta }\right]$$
where:$$\Delta ={\left[\left(3\varphi -1\right)k_{s}+\left(2-3\varphi \right)k_{f}\right]}^{2}+8k_{s}k_{f}$$

The subscripts s, f and nf refer to solid, fluid and nanofluid, respectively. Here, the parameter φ is the nanoparticle volume fraction, which is represented by the ratio of nanoparticle volume divided by the total volume of nanofluid. The 6% quantity of SiO_2_ nanoparticles is mixed with the Therminol^®^ VP-1 base fluid. The thermal properties of the SiO_2_ nanoparticles are listed in Table 4.

### 2.3. Description of Straight Conical Strip Inserts 

In the current work, straight conical strip inserts are attached to the central rod. The geometrical characteristics of the straight conical strip are optimized by changing different combinations of the strip height (H) and conical angle (θ), as shown in Figure 3. The straight conical strip attached to the central rod has a constant thickness (t) of 10 mm in the downstream direction, with the horizontal pitch distance (P) between conical strips set at 486 mm.

The conical angle varied, at 15°, 20°, 25° and 30°, whereas the strip height was increased from 15 mm–30 mm in increments of 5 mm. After obtaining the optimized single straight conical strip characteristics, extra cases were inspected by increasing the number of strips (N) to two, three and finally to four. The total number of examined cases was nineteen, in addition to the conventional absorber tube. All examined swirl-generator cases are presented in Figure 4. The first conical strip was located 10 mm (s) away from the absorber inlet, in order to allow a greater mass flow rate for the fluid entering the absorber tube. 

## 3. Numerical Modeling

All numerical simulations were performed using the open-source code Open Field Operation and Manipulation (Open-FOAM) with the conjugated heat transfer multi-region simple foam (chtMultiRegionSimpleFoam) solver. A second-order central differencing scheme was used for flow parameters, whereas the turbulent properties were discretized using the van Leer scheme (Van Leer [55]). The numerical treatment and discretization techniques utilized in this study have already been extensively benchmarked over a wide range of thermal-hydraulic and renewable energy configurations, see Afgan et al. [56], Filippone and Afgan [57], Guleren et al. [58], Han et al. [59], Wu et al. [60], Wu et al. [61], Wu et al. [62], Kahil et al. [63], Benhamadouche et al. [64], Nguyen et al. [65], Ejeh et al. [66], Ejeh et al. [67], Revell et al. [68], Ahmed et al. [69], Ahmed et al. [70], McNaughton et al. [71] and Abed et al. [72]. The boundary conditions applied in the current study are summarized below:

▪At the inlet boundary, uniform temperature and fixed velocity values were applied, along with the turbulence flow properties such as turbulent kinetic energy and specific dissipation rate based on the Re number and thermo-physical properties of heat transfer fluid. However, the pressure was applied as a zero gradient boundary condition.▪At the outlet boundary, the flow was fully developed. Thus, the velocity and temperature were applied as zero gradients, as well as all turbulence characteristics (specific dissipation rate and turbulent kinetic energy). The pressure is applied at fixed values with a zero boundary condition.▪At solid walls, the condition of no-slip is applied for velocity and a zero fixed value for turbulent kinetic energy, whereas a very large fixed value for the specific turbulence dissipation rate (ω) is used, as recommended by Menter [73], as given below: According to the following expression based on the distance between the first cell and solid wall $y_{1},\;\;$ the density and dynamic viscosity of the heat transfer fluid is given by:(30)$$\omega =\frac{60\mu }{0.075\rho y^{+}{}^{2}}$$

The non-uniform heat flux distribution on the external wall of the solar receiver uses the curve fitting equations extracted from the MCRT model as proposed by Kaloudis et al. [9] and Abed et al. [5]. This boundary condition acts by concentrating the absorbed solar energy over the lower part of the solar absorber more than the upper half, as shown in Figure 5.

### 3.1. Mesh Independence Study

The mesh independence study was performed for two types of geometries. The first geometry utilized a typical absorber tube (without swirl generators), with three different grids having different sized cells (fine grid with 2.4 million cells, medium grid with 1.8 million cells and coarse grid with 0.8 million cells). The second geometry was that of a single straight conical strip with the larger geometrical parameters (H = 30 mm and θ = 30°) inside the solar receiver. Three meshes were also examined for this configuration—fine mesh with 3.8 million cells, medium mesh with 3.4 million cells and coarse mesh with 2.4 million cells. All meshes were refined in near-wall regions in order to capture the flow physics for low Reynolds’s treatment and non-dimensional number $y^{+}\approx 1$. For the typical absorber mesh independence study, water with 320 K inlet temperature was used as the heat transfer fluid, whereas the Therminol VP-1 at 400 K inlet temperature was utilized in the second geometry. It is evident from Figure 6 and Figure 7 that the coarse meshes failed to predict the Nusselt number in the fully turbulent flow region at higher Reynolds numbers. However, the medium meshes in both cases were able to sufficiently predict the Nusselt numbers even in the fully turbulent flow regions over the entire range of Reynolds numbers. No further improvement was observed with the fine meshes for both the configurations; thus, the medium meshes were selected for further computations. For other conical strip geometries, the mesh size was adjusted based on the conical strip parameters, i.e., height (H) and angle (θ). Figure 6 and Figure 7 show the mesh independence study results for the typical receiver and the solar receiver with a straight conical strip, respectively.

### 3.2. Numerical Model Validation

The numerical results of a typical receiver are validated and compared with experimental and empirical correlations using two turbulence models—the LS k−ε model derived by Launder and Sharma [74] and the k−ω SST model proposed by Menter [73]. The mean Nusselt numbers, determined from the numerical results, were validated with the empirical correlation proposed by Gnielinski [46]. Water was used as the working fluid, with an inlet temperature of 320 K and a constant Prandtl (Pr) value of 3.77. The k−ω SST model was found to be more accurate as it determined the Nusselt number more accurately than the LS k−ε model, as shown in Figure 8. It can be observed that the k−ω SST model was in good agreement with the experimental data in all the turbulent regions.

Both turbulence models were used to predict the output temperature of the heat transfer fluid through the parabolic trough collector (without the glass envelope) under the non-uniform heat flux distribution. The external wall of the solar receiver used Syltherm 800 oil as the working fluid. Results were then compared to the experimental data of Dudley et al. [45]; comparisons are shown in Table 5. It can be seen from this table that the k−ω SST model performed better than the LS k−ε model.

The thermal efficiency of the parabolic trough collector without the glass envelope was also validated against the experimental data of Dudley et al. [45]. It can be observed from Figure 9 that the k−ω SST model once again gave better comparisons.

The computed friction factor f was also validated in the model with the empirical correlation proposed by Petukhov [47] over a wide range of Reynolds numbers. Findings revealed that the numerical predictions showed a good relationship with the experimental data, as shown in Figure 10.

Furthermore, the average Nusselt number predicted by the k−ω SST model was also validated with the experimental data of Pak and Cho [49] which utilized nanofluid γ-Al_2_O_3_-water with volume fractions of 1.34 and 2.78% under uniform heat flux. The numerically computed friction factor was also validated with the experimental measurements of Subramani et al. [75] under non-uniform heat flux distribution. All comparisons showed good agreements as shown in Figure 11 and Figure 12.

Based on all validations, it was concluded that the k−ω SST model performed better than the LS k−ε turbulence model. Thus, this turbulence model was utilized for all remaining numerical simulations.

## 4. Results and Discussions

### 4.1. Heat Transfer Performance

The use of swirl generators inside a parabolic trough collector system decreases the temperature gradient and enhances the convective heat transfer performance. This is achieved by absorbing more energy from the inner wall of the solar receiver and from the swirl generator. Figure 13 shows the behaviour of the Nusselt number of the solar receiver as a function of the Reynolds number for all angles and pitch heights. The figure also shows the effect of variation in the conical insert design, as well as the optimum case with and without nanofluid compared to the pure absorber (typical absorber). It is clearly evident from this figure that keeping the angle pitch constant and increasing the height pitch gradually (from 15 mm to 30 mm) leads to a gradual increase in the Nusselt number. This increase is due to two reasons: first, the increase in the contact surface area of the swirl generator with the working fluid and, second, added mixing of working fluid due to the presence of inserts.

Furthermore, increasing the angle pitch (from 15° to 30°) also enhanced the Nusselt number due to an increase in the contact surface area of the swirl generator. For the single conical insert configuration, the maximum Nusselt number was attained by the larger geometry, as shown in Figure 13d. These results are in line with the main findings of Mwesigye et al. [19]; Bellos et al. [25]; Bellos and Tzivanidis [20] and Liu et al. [31]. It can be observed in Figure 13e that the thermal energy absorption rate increases with an increase in the number of conical inserts, and the highest heat transfer performance was achieved by the four-strip conical insert. The presence of swirl generators inside the parabolic trough collector system increases the turbulence intensity of the working fluid. This leads to increased mixing with the swirl generators, as seen in the turbulent kinetic energy (TKE) distribution contours in Figure 14, which is beneficial for the augmentation of the heat transfer.

The optimum conical strip case was examined further in the presence of nanofluid (SiO_2_ mixed with the Therminol VP-1 (TO) using 6% volume fractions), as shown in Figure 13f. The introduction of nanofluid improved the thermal properties of the pure working fluid by enhancing the overall Prandtl number. This increased the thermal conductivity, enhanced the dynamic viscosity and reduced the specific heat capacity. It is thus obvious that the introduction of nanofluid helps in absorbing more energy from the solar receiver compared to the pure absorber configuration. It was thus concluded that combining inserts such as straight conical strips with nanofluids leads to significant augmentation in the Nusselt number, as also reported by Bellos and Tzivanidis [20].

Figure 13 also shows that by increasing the Reynolds number, the Nusselt number also increases. This was expected for several reasons—reduction in the thermal boundary layer thickness, lowering of the inside temperature of the solar receiver, and the decrease in the output working fluid temperature. The computed thermal enhancements in the Nusselt number of single and multiple conical cases with and without nanofluids at a Reynolds number of 10,000 are shown in Table 6.

### 4.2. Receiver Hydraulic Characteristics

The use of swirl generators results in an added pressure drop and a higher pumping power requirement compared to the plain absorber configuration. This is not desirable, as it leads to an increase in the investment cost of the system. In fact, large swirl generators have a significant effect on the working fluid pressure drop (due to the larger friction factor and blockage effects). Figure 15 presents the specific pressure drop (ΔP/L) change for all swirl generator configurations with and without nanofluids. It can be observed that at a constant Reynolds number, the pressure drop increases gradually with increasing pitch height and pitch angle, reaching the maximum value for the largest single insert geometry (H30mm, θ30°), as shown in Figure 15e. Similarly, a significant increase can be noted with an increase in the number of inserts. These results are in line with the findings of Mwesigye et al. [19], Xiangtao et al. [26] and Bellos and Tzivanidis [20]. The comparisons of thermal enhancement revealed that the pressure drop was more sensitive to the presence of the swirl generator than the Nusselt number. This was expected, as the insertion of swirl inserts adds to the friction and blockage.

Similar observations were made in the case of the nanofluid—an increase in the pressure drop compared to the pure working fluid configuration, due to the change in fluid characteristics such as dynamic viscosity and fluid density. Thus, the pumping requirement increases for the nanofluids configurations. Increasing the working Reynolds number also caused a considerable increase in the pressure drop due to the increase in the flow turbulence which requires larger pressure drop to force workings moving through the solar absorber tube. It is worth mentioning here that the current study combined the swirl generators with nanofluid, which resulted in an added pumping requirement, as shown in Figure 15. The main aim here is heat transfer augmentation, and power requirements are not a constraint as far as this study is concerned. However, from a practical installation perspective one cannot ignore the power input requirement.

For single and multiple conical inserts with and without nanofluids, the specific pressure drop changes are listed in Table 7. It can be concluded that the increase in the specific pressure drop due to the presence of the swirl generators was on average 11.78% more than that produced by the introduction of nanofluids. 

### 4.3. Performance Evaluation Criterion

The performance evaluation criterion (PEC) can be used to assess the thermal and hydraulic performances of any passive or active technology that is integrated in a typical solar receiver system. This parameter is based on the relative comparison of the Nu number and friction factor of the enhancement configuration compared to the base case, as shown in the Equation (10).

If the PEC  is greater than one, it means that both the Nu number and pressure drop produced by the applied technology have been enhanced and increased. However, if the PEC is less than one, it means that the pressure drop increase is larger than the Nu number enhancement. The PEC behaviour of all insert configurations is presented in Figure 16. It can be observed that the levels of enhancement in the thermal and hydraulic performances fluctuate and change for all configurations, with the general trend being that the PEC increases with an increase in the pitch angle and decreases with an increase in the number of inserts. However, for the combined case of nanofluid with inserts, the gain in the Nusselt number is significantly larger that the increase in the pressure drop and thus the PEC for this optimal case stands at 1.423.

### 4.4. Thermal Losses

The introduction of swirl generators inside a parabolic trough collector significantly reduces the thermal losses as they increase the absorption of thermal energy from the inner wall surface of the solar receiver. This results in extracting more thermal energy from the external wall of the solar receiver, thereby reducing the external wall temperature. Thus, increasing the surface area of swirl generators leads to an increase in the absorption of thermal energy, leading to a gradual reduction in the convection and radiation thermal losses over the external walls of the solar receiver. Figure 17 presents the thermal losses for all configurations with and without nanofluid.

It can be observed from Figure 17a–d that by keeping the pitch angle constant, the thermal losses can be reduced simply by increasing the pitch height. Similarly, increasing the pitch angle also reduces the thermal losses. These losses can further be reduced by increasing the number of conical inserts, as show in Figure 17e. Similar conclusions were also drawn by a number of previous studies—Cheng et al. [16]; Bellos and Tzivanidis [20]; Liu et al. [31]; Arshad et al. [28].

An additional technique examined in this study was to analyse the mixing of 6% volume fraction of SiO_2_ nanoparticles in Therminol VP-1. This technique resulted in the development of a new working fluid which has better thermal properties than that of the pure working fluid, thereby reducing the thermal losses further, as shown in Figure 17f. Finally, the combination of swirl inserts with nanofluids leads to the most optimum case, in which the reduction of losses is about 23.7%. It is prudent to mention here that overall, the reduction in losses was found to be more sensitive to the increase in the number of inserts than to the use of the nanofluids. Another important feature that can be noted from Figure 17 is that the thermal losses were lower for the high Reynolds number cases. This is due to the enhancement of turbulence and the increase in the flow path, thus the working fluid has an opportunity to collect more thermal energy than that in the case of pure absorber. 

Finally, the reductions in thermal losses of single and multiple swirl generators with and without nanofluids are presented in Table 8, at a Reynolds number of 10,000.

### 4.5. Overall Collector Efficiency

The overall thermal efficiency, considering the effect of pumping power required for the conical insert configuration, can be used for the evaluation of the system, as shown in Figure 18. It is evident from Figure 18a–e that the overall thermal efficiency increases with both the pitch angle and the pitch height for the single canonical insert. With the introduction of additional inserts, this efficiency further improves, as also reported by Mwesigye et al. [18]; Mwesigye et al. [19]; Bellos and Tzivanidis [20]; Liu et al. [31]; and Arshad et al. [28]. Once again, when the two techniques were combined—that is, swirl inserts with nanofluid—the efficiency increase was the highest, as shown in Figure 18f.

The enhancement in the overall thermal efficiency is due to several factors, but it is mainly due to an increase in absorption, which leads to improved convective heat transfer behavior and reduced the convection and radiation thermal losses. This results in higher output working fluid temperatures and useful heat energy gain. Thus, one can conclude that by reducing the solar receiver wall temperature gradient, the convection and radiation thermal losses can be reduced. It can be further noted that the overall thermal efficiency was better at lower Reynolds numbers. This is because at high Reynolds numbers, the gain in the convection heat transfer is not enough to overcome the pressure drop and thermal losses at these high flow velocities.

The main improvement in the overall thermal efficiency for single and multiple swirl generators with and without nanofluid are given in Table 9 for a Reynolds number of 30,000.

### 4.6. Thermal Exergy Efficiency 

To assess the actual effect of introducing swirl generators on the maximum net possible work production of the parabolic trough collector system, the thermal exergy efficiency was considered for assessment. A comparison of thermal exergy efficiency for all considered cases is presented in Figure 19. Once again it can be observed that the trends of thermal exergy efficiency are very similar to the overall collector efficiency, i.e., it increases with an increase in the pitch angle, an increase in the pitch height, and with an increase in the of number of inserts, as also reported by Bellos and Tzivanidis [20] and Bellos et al. [27].

Finally, it can be seen in Figure 19f that even though the addition of nanofluid to the insert configuration improves the exergy efficiency (in the optimal case of four inserts with nanofluid), the gain by adding only the nanofluid was not as significant as the gain in the case of inserts alone. The main improvements in the thermal exergy efficiency for single and multiple swirl generators with and without nanofluids are shown in Table 10 at a Reynolds number of 30,000.

## 5. Conclusions

Improving the overall efficiency of parabolic trough collectors can lead to enhancements in sustainable energy extraction and the reduction of greenhouse gas emissions. This goal was accomplished using three approaches. The first approach was to investigate the optimum configuration for straight conical strip inserts, which were attached to a central rod with varying parameters of pitch height, angle pitch, and the number of strip inserts. The second approach was to introduce 6% SiO_2_ nanoparticles mixed with Therminol VP-1 at an inlet temperature of 400 K. Finally, the third approach was the combination of the optimized configuration swirl generator insert with nanofluids. The evaluation of these approaches was based on the Nusselt number, pressure drop, performance evaluation criterion, thermal losses, overall thermal efficiency and thermal exergy efficiency. The main findings of this work are summarized below:▪The use of straight conical strips enhanced the Nusselt number by 47.13% whereas using nanofluids alone improved the Nusselt number by 15.57%. However, an improvement of 57.48% was observed for the Nusselt number by combining the swirl inserts and nanofluids. This combination also resulted in the maximum reduction of thermal losses by 23.7%.▪The improvement in the Nusselt number comes at the expense of an increase in the pressure drop. Swirl generators and nanofluids alone increased the pressure drop by 258.42% and 231.18%, respectively. However, when these were combined, the pressure drop reached as high as 348.03%.▪All the examined cases showed different levels of enhancements in the overall thermal efficiency and thermal exergy efficiency. For the combined case of nanofluid with optimum swirl generator configuration, the overall thermal efficiency improved by 14.62% and the thermal exergy efficiency increased by 14.47%.▪The gain in improvement of all tested parameters was found to be more sensitive to the insert geometry and the number of inserts as opposed to the nanofluids. Thus, it can be concluded that swirl inserts are a better candidate than nanofluids for thermal performance improvement in PTC systems.

## Figures and Tables

**Figure 1 nanomaterials-11-00853-f001:**
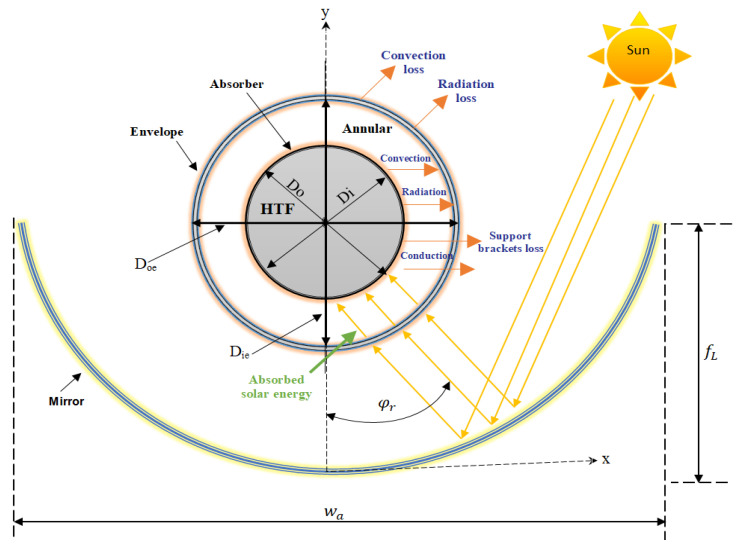
Schematic diagram of the parabolic trough and its receiver.

**Figure 2 nanomaterials-11-00853-f002:**
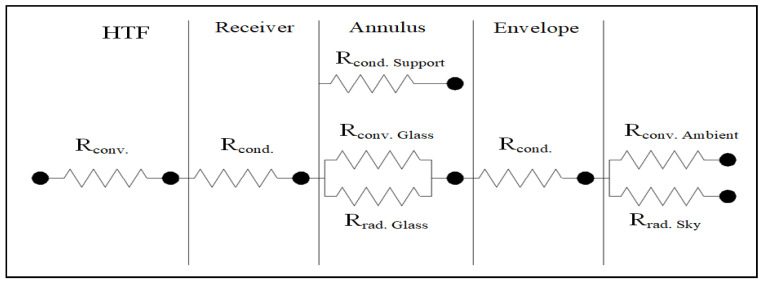
The parabolic trough collector (PTC) thermal resistance model. HTF, heat transfer fluid.

**Figure 3 nanomaterials-11-00853-f003:**
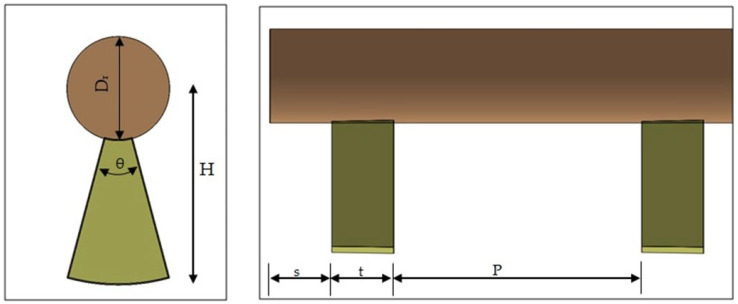
The conical strip geometery examined in the current study.

**Figure 4 nanomaterials-11-00853-f004:**
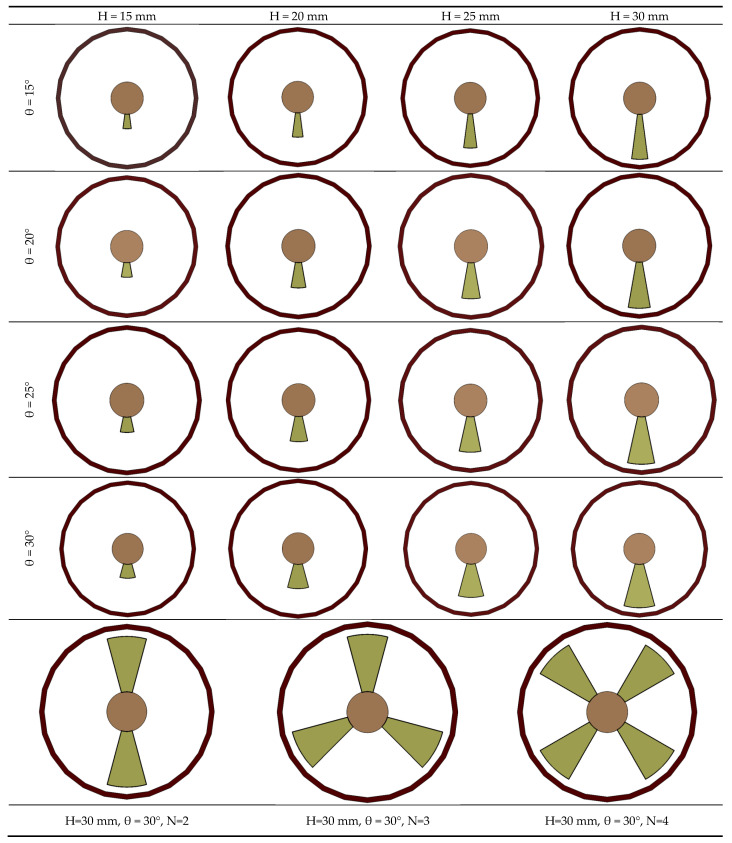
Various configurations of the examined straight conical strips in the current study.

**Figure 5 nanomaterials-11-00853-f005:**
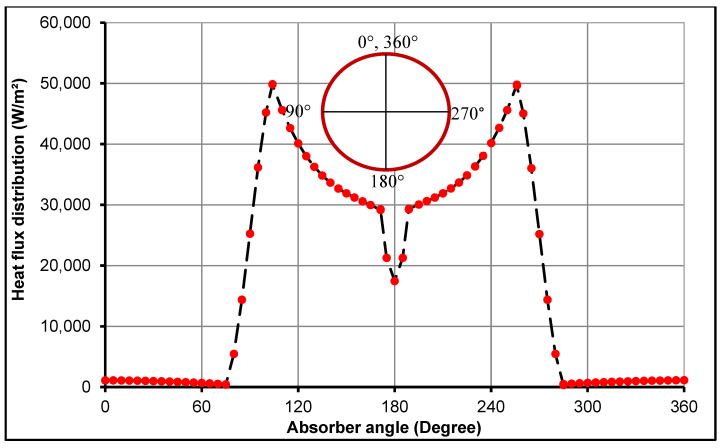
Non-uniform heat flux profile on the outer surface of the absorber receiver.

**Figure 6 nanomaterials-11-00853-f006:**
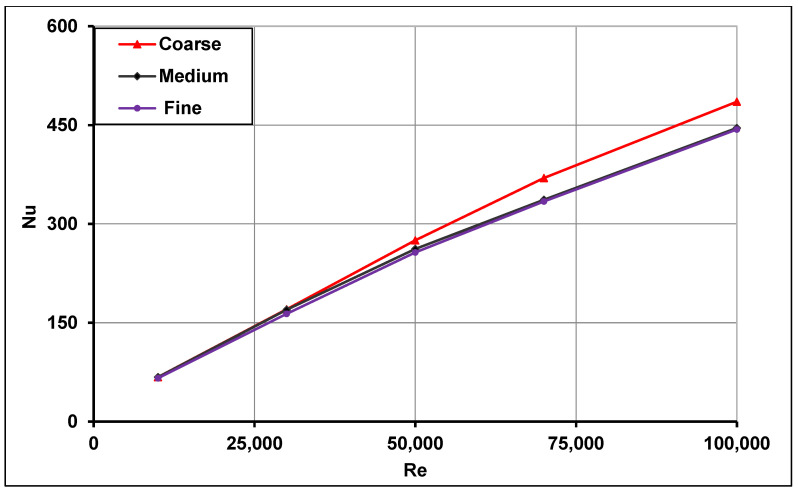
Mesh independence study for a typical receiver without inserts.

**Figure 7 nanomaterials-11-00853-f007:**
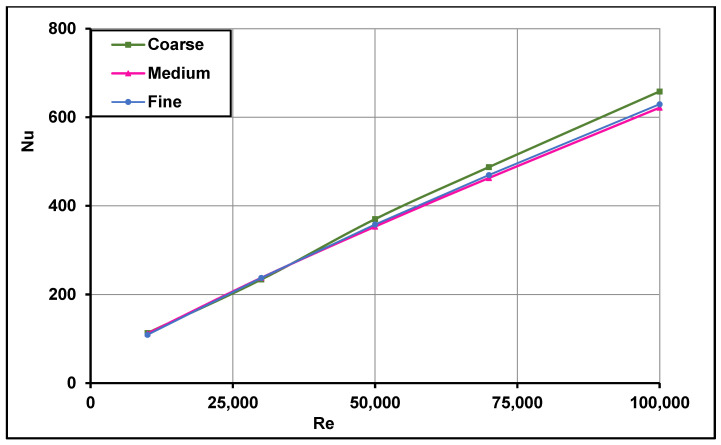
Mesh independence study for the solar receiver with a single straight conical strip insert.

**Figure 8 nanomaterials-11-00853-f008:**
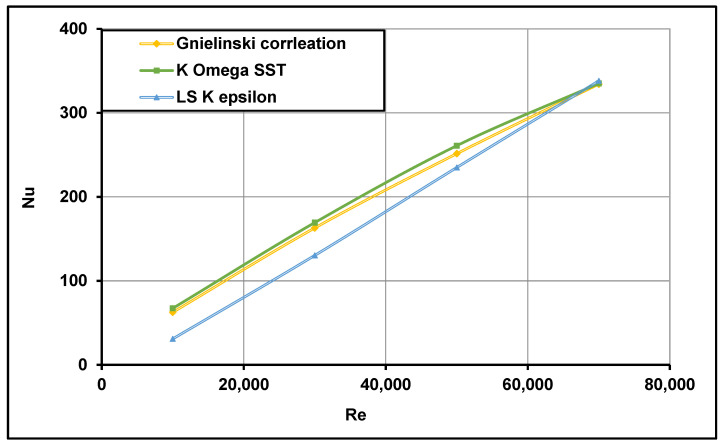
Computed average Nu number comparisons with measurements of Gnielinski [46].

**Figure 9 nanomaterials-11-00853-f009:**
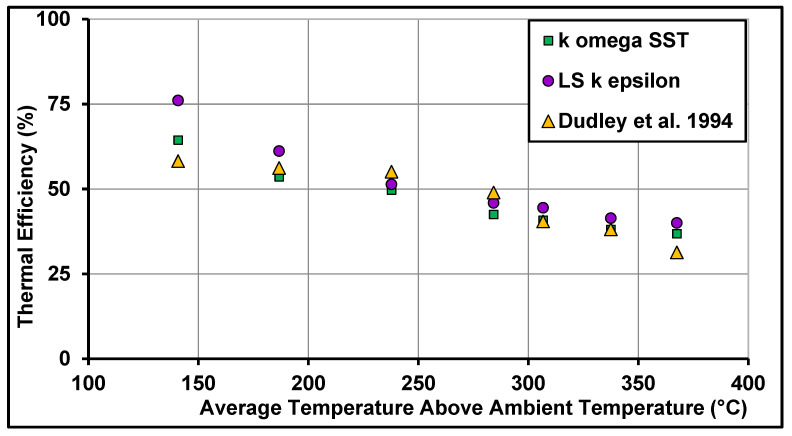
Computed thermal efficiency compassions against measurements of Dudley et al. [45].

**Figure 10 nanomaterials-11-00853-f010:**
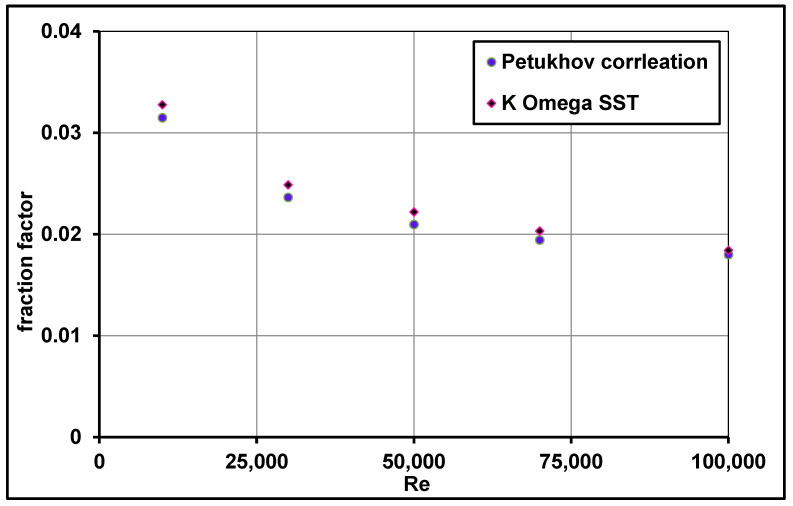
Computed friction factor comparisons with the empirical correlation of Petukhov [47].

**Figure 11 nanomaterials-11-00853-f011:**
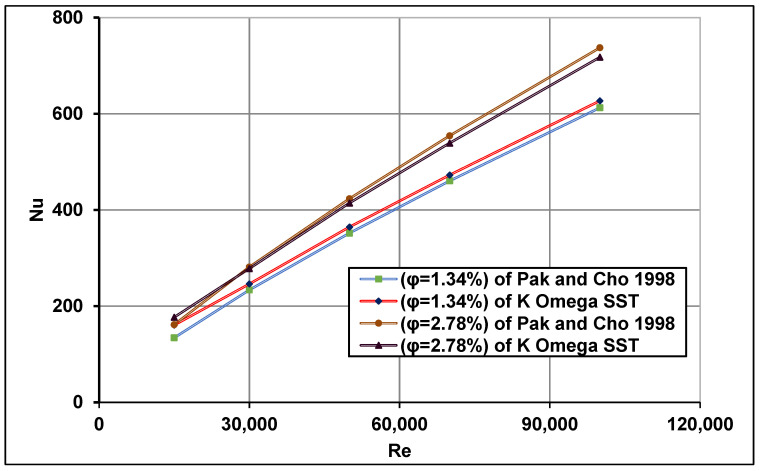
Computed average Nu number comparisons with measurements of Pak and Cho [49].

**Figure 12 nanomaterials-11-00853-f012:**
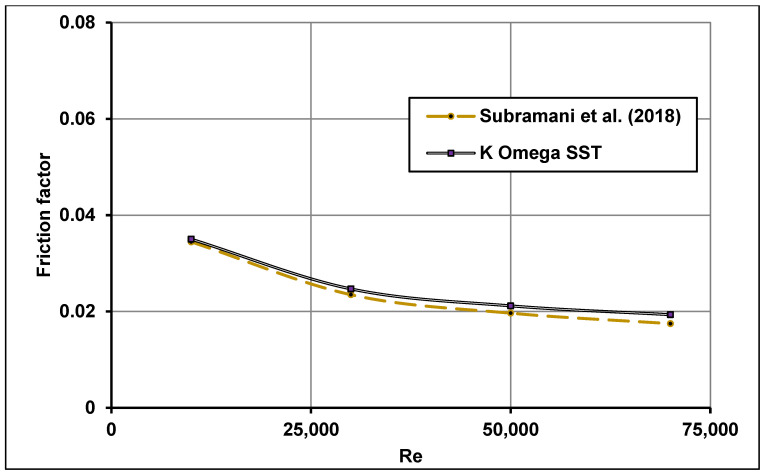
Computed friction factor comparisons with measurements of Subramani et al. [75].

**Figure 13 nanomaterials-11-00853-f013:**
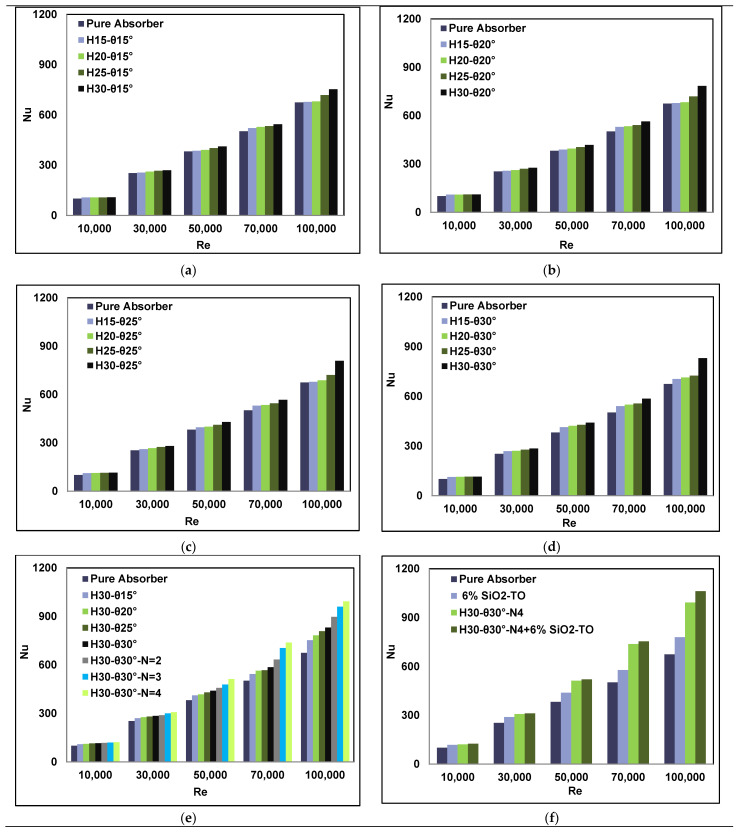
Effects of swirl generators geometries with/without nanofluids on heat transfer performance. (**a**) Pitch height variation with θ= 15°, (**b**) Pitch height variation with θ= 20°, (**c**) Pitch height variation with θ= 25°, (**d**) Pitch height variation with θ= 30°, (**e**) Pitch angle (θ) and number of inserts (N) variation with fixed pitch height, (**f**) Optimal configuration with and without nanofluids.

**Figure 14 nanomaterials-11-00853-f014:**
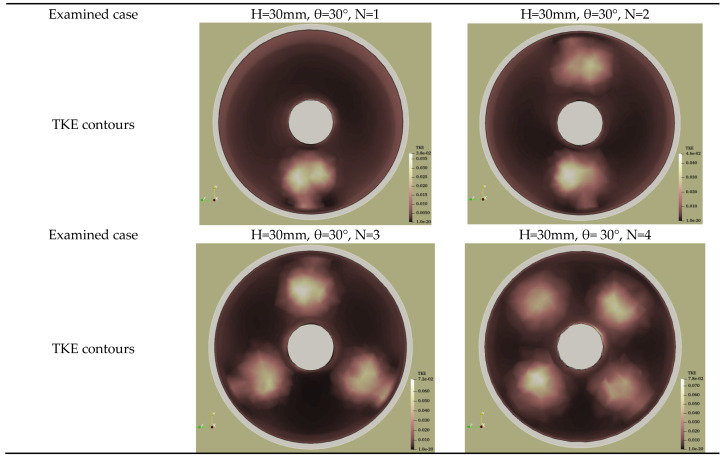
Effect of number of strip-inserts on the turbulent kinetic energy (TKE) distributions.

**Figure 15 nanomaterials-11-00853-f015:**
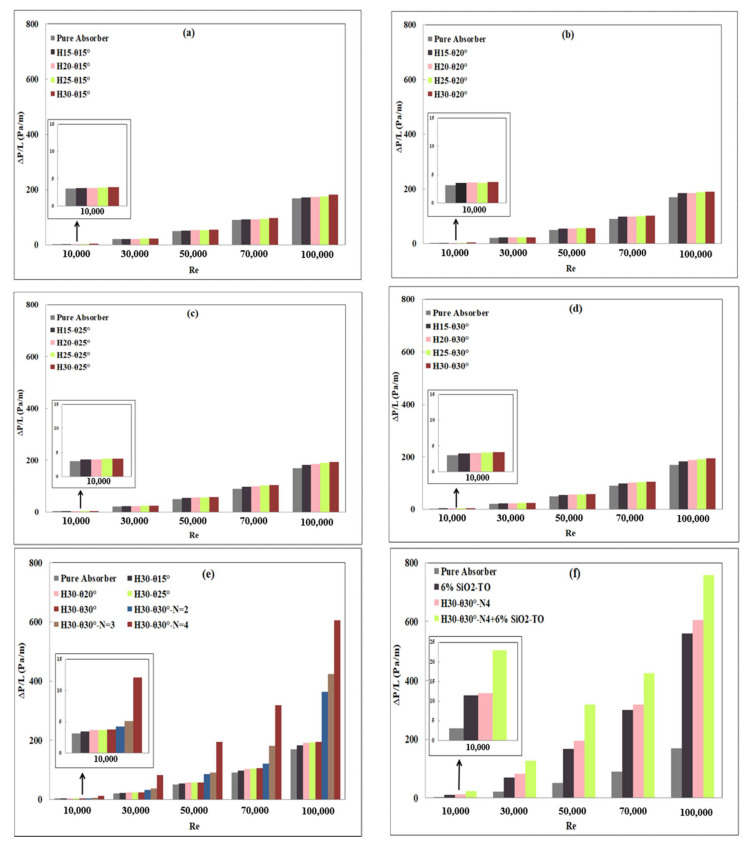
Pressure drop changes in the presence of swirl generators with and without nanofluid. (**a**) Pitch height variation with θ= 15°, (**b**) Pitch height variation with θ= 20°, (**c**) Pitch height variation with θ= 25°, (**d**) Pitch height variation with θ= 30°, (**e**) Pitch angle (θ) and number of inserts (N) variation with fixed pitch height, (**f**) Optimal configuration with and without nanofluids.

**Figure 16 nanomaterials-11-00853-f016:**
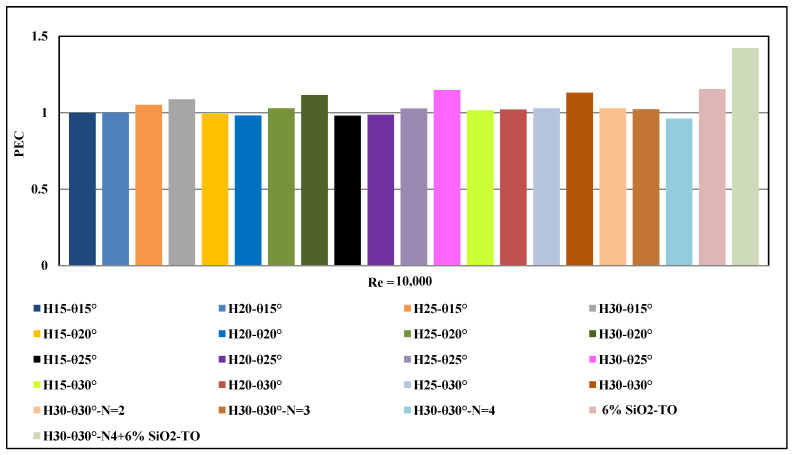
Effect of configuration changes on the performance evaluation criterion (PEC).

**Figure 17 nanomaterials-11-00853-f017:**
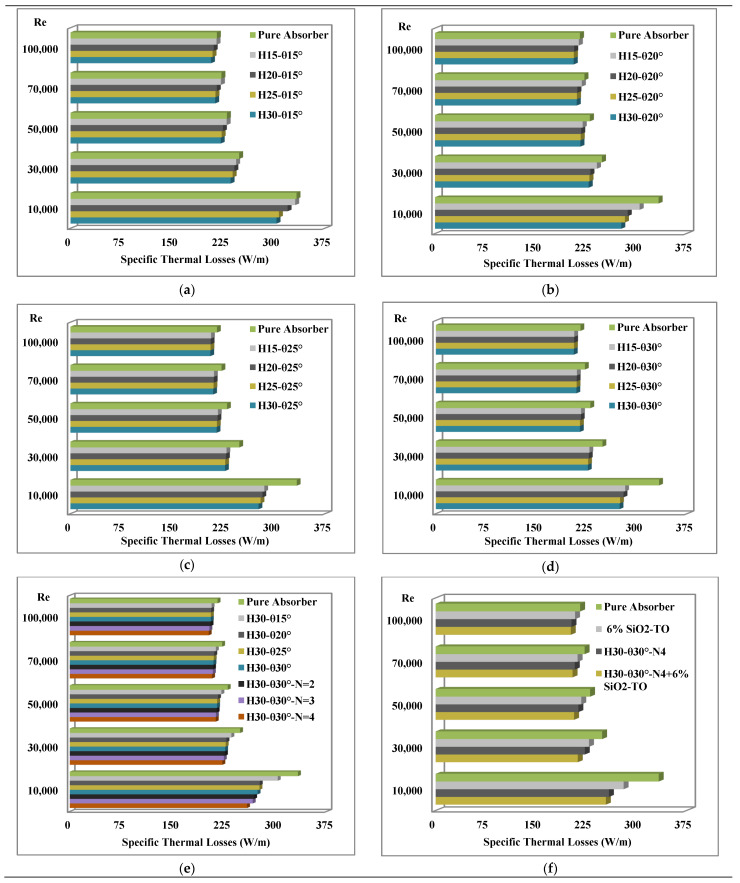
Comparison of thermal losses for all considered configurations. (**a**) Pitch height variation with θ= 15°, (**b**) Pitch height variation with θ= 20°, (**c**) Pitch height variation with θ= 25°, (**d**) Pitch height variation with θ= 30°, (**e**) Pitch angle (θ) and number of inserts (N) variation with fixed pitch height, (**f**) Optimal configuration with and without nanofluids.

**Figure 18 nanomaterials-11-00853-f018:**
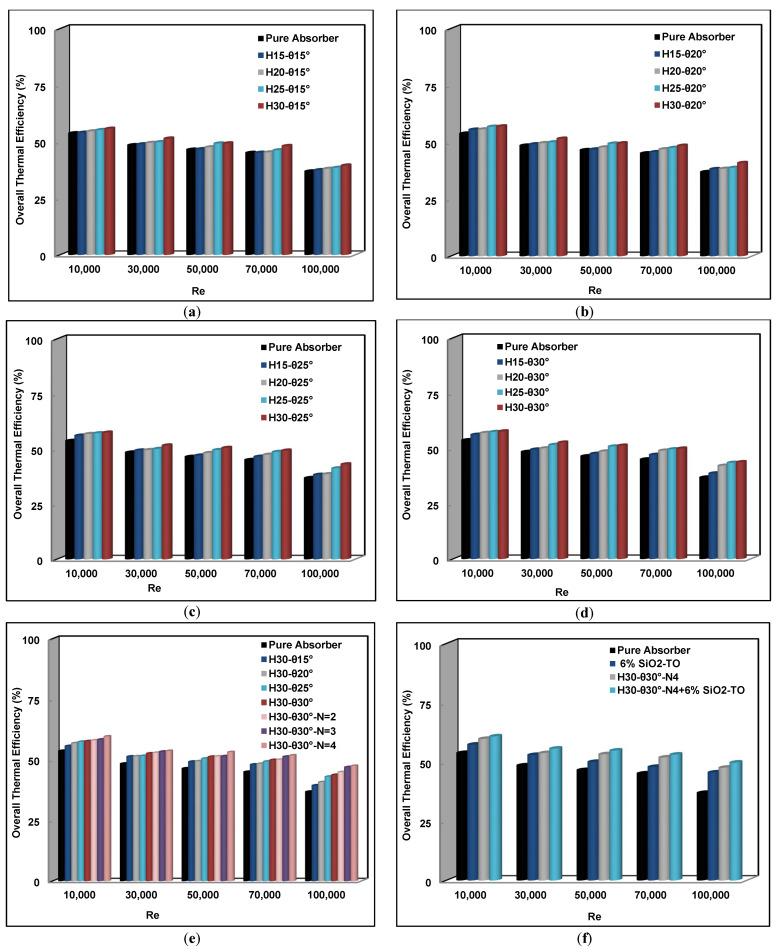
Overall thermal efficiency comparisons for all considered cases. (**a**) Pitch height variation with θ= 15°, (**b**) Pitch height variation with θ= 20°, (**c**) Pitch height variation with θ= 25°, (**d**) Pitch height variation with θ= 30°, (**e**) Pitch angle (θ) and number of inserts (N) variation with fixed pitch height, (**f**) Optimal configuration with and without nanofluids.

**Figure 19 nanomaterials-11-00853-f019:**
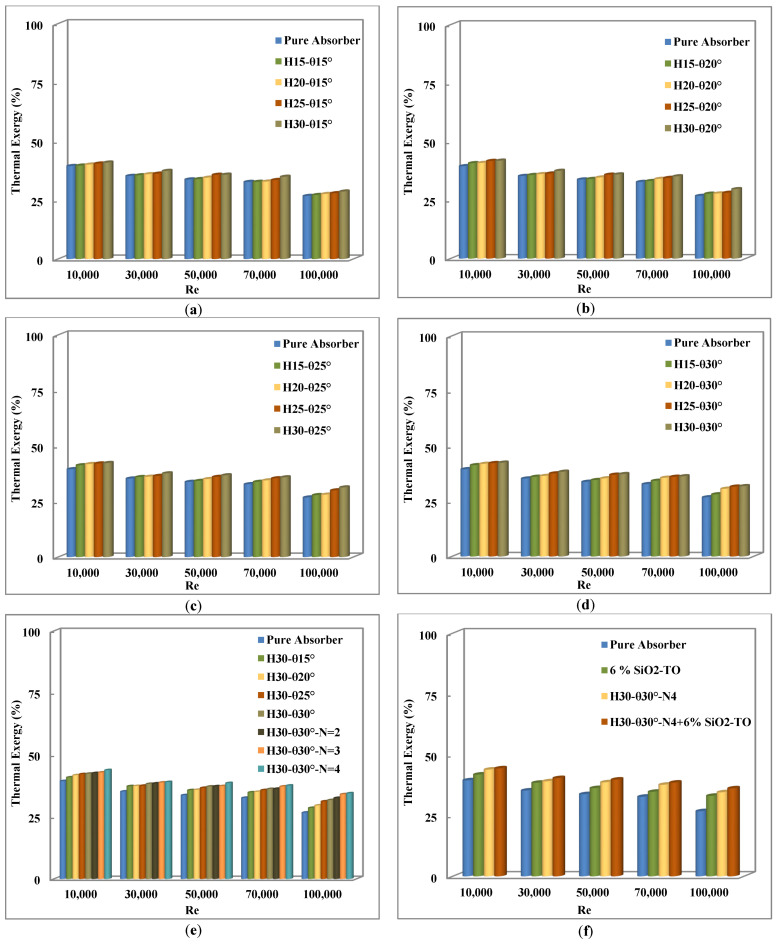
Comparison of thermal exergy efficiency for all considered cases. (**a**) Pitch height variation with θ= 15°, (**b**) Pitch height variation with θ= 20°, (**c**) Pitch height variation with θ= 25°, (**d**) Pitch height variation with θ= 30°, (**e**) Pitch angle (θ) and number of inserts (N) variation with fixed pitch height, (**f**) Optimal configuration with and without nanofluids.

**Table 1 nanomaterials-11-00853-t001:** Notable studies on the effects of different insert shapes in the literature.

Ref.	Technique Used	Achievements
Mwesigye et al. [19]	Twisted tape type with wall-detached	The heat transfer coefficient and friction factor increased from 1.05 to 2.69 and 1.6 to 14.5, respectively. Moreover, due to the presence of twisted tapes, the difference in the temperature of the tube in the circumferential direction was reduced by 68%. However, the thermal efficiency increased by 5%–10% at a twist ratio larger than 1 and entropy generation was reduced by 58.8%.
Bellos and Tzivanidis [20]	Star flow	A 1% increase in the thermal efficiency was recorded.
Song et al. [21]	Helical screw tape inserts with core rod	The maximum temperature was reduced by six times and heat losses by three times. However, the pressure drop increased by four times in the typical receiver and 23 times for the helical screw tape inserts.
Chang et al. [22]	Twisted tapes	The heat transfer and friction factor improved significantly by decreasing clearance and twisted ratios. This decrease caused the Nusselt number to increase by 2.9 times and the friction factor by 1.24, much larger than the smooth receiver.
Zheng et al. [23]	Dimpled twisted tapes	Heat transfer showed a significant increase with inserts. However, the dimple side provided better heat transfer performance than the protrusion side, showing that the heat transfer coefficient increased by 25.53%, with a 29.1% reduction in the average entropy generation.
Zhu et al. [24]	Wavy tape	The heat losses and entropy generation were reduced by 17.5%–33.1% and 30.2%–81.8% respectively.
Bellos et al. [25]	Twelve-fin geometries	The large length and thickness provided better thermal and hydraulic performances. However, the length of the fins was more important than the thickness, in which the enhancement index was found to be 1.483.
Xiangtao et al. [26]	Pin fins arrays	Nusselt number increased by 9% and thermal performance factor increased up to 12% with the optimum case of 8 mm as the fin diameter at a Reynolds number of 4036.
Bellos et al. [27]	Multiple cylindrical inserts	The thermal losses were reduced by 5.63% with a 26.88% enhancement in the heat transfer coefficient.
Arshad et al. [28]	Internal toroidal rings	Nusselt number and thermal efficiency increased by 3.74% and 2.33 times, respectively.
Rawani et al. [29]	Serrated twisted tape and metal foam	5% and 3% increase in the thermal efficiency when using serrated twisted tape and metal foam, respectively.
Bellos et al. [30]	Eccentric inserts	A 1% increase in the thermal efficiency was reported.
Liu et al. [31]	Inclined conical strip inserts	A 5% enhancement in the thermal efficiency was recorded.
Yılmaz et al. [32]	Wire coil inserts	183% improvement in heat transfer performance with a 0.4% increment in the thermal efficiency was reported.
Kumar and Reddy [33]	Metal foams	3.71% increase in the net energy efficiency and 2.32% in the exergy efficiency. However, the temperature difference was reduced from 47% to 72%, compared to the plain absorber.
Suresh et al. [34]	Modified twisted tapes	The Nusselt number was enhanced by 5%–40%, 11%–101% and 7%–77% in typical twisted tape, twisted tape with attached rings and twisted tapes with modified attached rings, respectively.
Xiao et al. [35]	Curved-twisted baffles	0.52% and 0.22% increase in the overall efficiency and exergy efficiency, respectively.
Bellos et al. [27]	Internal fins and nanofluids	0.76% enhancements in the thermal efficiency after using 6% CuO-thermal oil nanofluids. They also reported that the thermal efficiency increased by 1.1% when using internal fins. Moreover, the use of nanofluid and internal fins increased the thermal efficiency by 1.54%.

**Table 2 nanomaterials-11-00853-t002:** The PTC model parameters used in the current work.

Property	Value	Property	Value
Inner diameter of absorber tube, D_i_	0.066 m	Focal length, $f_{L}$	1.84 m
Outer diameter of absorber tube, D_o_	0.07 m	Aperture width, wa	8.0 m
Solar receiver length, L	4.0 m	Rim angle, $\varphi_{r}$	95°
Solar beam irradiation, G_b_	1000 W/m^2^	Wind speed, V	0.5 m/s
Absorber specific heat capacity	512 J/kg·K	Absorber density	8050 kg/m^3^
Absorber thermal conductivity	17.3 W/m·K	Ambient temperature, Tam	300 K

**Table 3 nanomaterials-11-00853-t003:** Thermal properties of Therminol® VP-1 heat transfer fluid, used in the present study.

T (K)	μ (Pa·s)	ρ (kg/m^3^)	C_p_ (J/kg·k)	k (W/m·k)	Pr	Reference
400	0.000732	975.8	1850.5	0.1243	10.89	Abed et al. [5]

**Table 4 nanomaterials-11-00853-t004:** Thermal properties of the nanoparticles considered in the present work.

Name	ρ (kg/m^3^)	C_p_ (J/kg·k)	k (W/m·k)	Particle Type	Size (nm)	Reference
SiO_2_	2200	765	1.4	Sphere	20	Bellos and Tzivanidis [53], Al-damook et al. [54]

**Table 5 nanomaterials-11-00853-t005:** Comparisons of the output temperature of the working fluid.

Dudley et al. [45]					k−ω SST Model		LS k−ε Model	
$V^{˙}\;(L/\min)$	Pr	G_b_ (W/m^2^)	T_in_ (°C)	T_out_ (°C)	T_out_ (°C)	Deviation (%)	T_out_ (°C)	Deviation (%)
48.4	27.92	801.3	151.7	166.2	168.337	−1.286	171.347	−3.097
49.8	19.93	888.6	198.2	215.5	213.294	1.024	215.425	0.035
51.1	12.02	920.5	301	314.2	313.809	0.124	314.833	−0.201
55.6	11.37	929.4	313.8	324.8	325.340	−0.166	326.375	−0.485
55.8	8.95	940.4	384	395	395.446	−0.113	396.435	−0.363
50.9	14.85	935.7	252.1	268	266.868	0.422	267.360	0.239
39.8	42.7	817.5	101	120.8	127.859	−5.844	136.211	−12.757
50.1	19.32	854.5	203.1	219.2	217.292	0.870	219.364	−0.075
50	19.29	867.6	203.4	219.6	217.825	0.808	219.210	0.178
48.2	42.9	922	100.8	121.1	125.409	−3.558	131.945	−8.955
51.6	9.55	927.6	354.4	367.8	366.507	0.352	367.572	0.062
	Mean Deviations (%):				−0.670		−2.311	

**Table 6 nanomaterials-11-00853-t006:** Thermal enhancement in the Nu number with single and multiple conical inserts, with and without nanofluid at Re = 10,000.

Case	Nu Number	Enhancement (%)
Pure absorber	674.83	-
H30-θ30°-N1	831.25	23.17
H30-θ30°-N2	896.55	32.85
H30-θ30°-N3	959.88	42.24
H30-θ30°-N4	992.89	47.13
6% SiO_2_-TO	779.95	15.57
H30-θ30°-N4+6% SiO_2_-TO	1062.78	57.4

**Table 7 nanomaterials-11-00853-t007:** Hydraulic enhancement in t pressure drop for single and multiple conical inserts with and without nanofluid.

Case	ΔP/L	Enhancement (%)
Pure absorber	169.09	-
H30-θ30°-N1	218.18	29.03
H30-θ30°-N2	363.63	115.053
H30-θ30°-N3	454.54	168.81
H30-θ30°-N4	606.06	258.42
6% SiO_2_-TO	560	231.18
H30-θ30°-N4+6% SiO_2_-TO	757.57	348.03

**Table 8 nanomaterials-11-00853-t008:** Thermal loss reductions for single and multiple conical insert cases with and without nanofluid at Re = 10,000.

Case	Specific Thermal Losses (W/m)	Reduction (%)
Pure absorber	333.26	−
H30-θ30°-N1	274.29	−17.6
H30-θ30°-N2	269.21	−19.2
H30-θ30°-N3	267.206	−19.82
H30-θ30°-N4	258.88	−22.3
6% SiO_2_-TO	280.85	−15.72
H30-θ30°-N4+6% SiO_2_-TO	254.3	−23.7

**Table 9 nanomaterials-11-00853-t009:** Overall thermal efficiency enhancement for single and multiple conical insert cases with and without nanofluid at Re = 30,000.

Case	Overall Thermal Efficiency (%)	Improvement (%)
Pure absorber	48.61	-
H30-θ30°-N1	52.81	8.637
H30-θ30°-N2	53.08	9.184
H30-θ30°-N3	53.57	10.19
H30-θ30°-N4	53.84	10.74
6% SiO_2_-TO	53.05	9.18
H30-θ30°-N4+6% SiO_2_-TO	55.72	14.62

**Table 10 nanomaterials-11-00853-t010:** Thermal exergy efficiency enhancement for single and multiple conical insert cases with and without nanofluid at Re = 30,000.

Case	Thermal Exergy Efficiency (%)	Improvement (%)
Pure absorber	35.36	-
H30-θ30°-N1	38.43	8.67
H30-θ30°-N2	38.63	9.22
H30-θ30°-N3	38.98	10.24
H30-θ30°-N4	39.18	10.79
6% SiO_2_-TO	38.557	9.02
H30-θ30°-N4+6% SiO_2_-TO	40.48	14.4

## Data Availability

Data is contained within the article.

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
