# Peer review of "Thermal-Hydraulic Analysis of Parabolic Trough Collectors Using Straight Conical Strip Inserts with Nanofluids"

_nanomaterials, 2021, doi:10.3390/nano11040853_

Round 1
Reviewer 1 Report
Review:
Thermal-hydraulic Analysis of Parabolic Trough Collectors using Straight Conical Strip Inserts with Nanofluids
Line 140: Please specify the type, size and morhology of the SiO2
Table 3: Whether it makes sense to consider the stated density, given that we know that these properties are most critical for nanomaterials. Please explain why you did not take into account the bulk density for SiO2 which has the value < 0.1 g/cm3 - what would the deviation of the presented calculations be?? Please explain.
Please explain how these nano SiO2 are made?

Author Response
Suggested corrections and/or reply to reviewer comments
Manuscript ID: nanomaterials-1142429
Title: Thermal-hydraulic Analysis of Parabolic Trough Collectors using straight Conical Strip Inserts with Nanofluids
Authors: Nabeel Abed, Imran Afgan, Hector Iacovides, Andrea Cioncolini, Ilyas Khurshid, Adel Nasser
Reviewer 1
|
Reviewers Comments/Suggestions |
Reply to the reviewer Please note: Corrections/response are shown as red texts in the new version of the manuscript |
Changes made Page No. |
|||||||||||||||||||||
|
Line 140: Please specify the type, size and morphology of the SiO2 |
The details of nanoparticles are modified in the Table 4 (it was Table 3 in the previous version) as follows:
|
13 |
|||||||||||||||||||||
|
Table 3: Whether it makes sense to consider the stated density, given that we know that these properties are most critical for nanomaterials. Please explain why you did not take into account the bulk density for SiO2 which has the value < 0.1 g/cm3‐what would the deviation of the presented calculations be?? Please explain |
The authors appreciate the reviewer’s concerns. The range of SiO2 density is between 2170-2650 kg/m3 based on the following source: (https://www.azom.com/properties.aspx?ArticleID=1114) However, this type of nanoparticles is mixed with a working fluid has a density lighter than the nanoparticles. Therefore, for proper single-phase calculations, the mixture of solid particle and base fluid should be both taken into account which gives a density heavier than the base fluid and lighter than that of the nanoparticles alone. |
|
|||||||||||||||||||||
|
Please explain how these nano SiO2 are made? |
The authors appreciate the reviewer’s concerns. The following paragraph has been added as requested. In the experimental work, the preparation of nanofluids can be performed by different methods. In general, there are two main methods that have been proposed in the literature; the one-step approach and the two-step approach. In the first approach, the formations of both nanoparticles and base fluid are combined together. However, if the nanoparticles are already prepared and are then mixed with the base fluids, the procedure is considered as the two-steps approach. For numerical simulations, the application of nanofluids can either be a single-phase process or two-phase process using different models proposed in the literature which depend on the material/fluid properties such as; density, specific heat capacity, thermal conductivity and dynamic viscosity, Michaelides (2016).
|
12 |
|||||||||||||||||||||

Reviewer 2 Report
Manuscript ID: nanomaterials-1142429
Type of manuscript: Article
Title: Thermal-hydraulic Analysis of Parabolic Trough Collectors using Straight Conical Strip Inserts with Nanofluids
Authors: Nabeel Abed, Imran Afgan, Hector Iacovides, Andrea Cioncolini, Ilyas Khurshid, Adel Nasser
Presented manuscript is focused on the use of swirl generator technique inside parabolic trough collectors as a common passive approach to improve the thermal-hydraulic performances. A detailed literature survey are well made.
The literature data and experimental part are well described. The results were discussed and the conclusions were also sound. I recommend publication of this work in Nanomaterials after a minor revision.
(1) The Abstract must be concise. Please reduce the Abstract part eliminating the details and summarized it.
(2) In the Introduction part maybe the discussion from page 4, line 81 to page 5, line 131 can be summarized in a Table with advantages and disavantages.
(3) In The Results and Discussion part the following paragraph can be removed: "The numerical results determined in the current study are presented and discussed in this section. [..]This section includes thermal energy enhancement, friction factor, pressure drop, performance evaluation criterion, thermal losses, overall thermal efficiency and energy efficiency."
(4) From Conclusion part the following text must be removed : " The objective of this study was to improve the 654 thermal and thermodynamic performances of the parabolic trough collector systems". The phrase must be reformulated as a conclusion. Conclusions should be concise and summarize all the important findings of the manuscript.
Author Response
Suggested corrections and/or reply to reviewer comments
Manuscript ID: nanomaterials-1142429
Title: Thermal-hydraulic Analysis of Parabolic Trough Collectors using straight Conical Strip Inserts with Nanofluids
Authors: Nabeel Abed, Imran Afgan, Hector Iacovides, Andrea Cioncolini, Ilyas Khurshid, Adel Nasser
Reviewer 2
|
Reviewers Comments/Suggestions |
Reply to the reviewer Corrections/response are shown as blue texts in the new version of the manuscript |
Changes made Page No. |
|
(1) The Abstract must be concise. Please reduce the Abstract part eliminating the details and summarized it. |
The authors appreciate the reviewer’s concerns. The abstract has been summarized as follows: This study numerically investigates the effect of swirl inserts with and without nanofluids over a range of Reynolds numbers for parabolic trough collectors with non-uniform heating. Three approaches were utilized to enhance the thermal-hydraulic performance; variation of geometrical properties of a single canonical insert to find the optimized shape, the use of nanofluids and effect of both previous approaches; swirl generators and nanofluids. Results reveal that by using the straight conical strips alone enhanced the Nusselt number by 47.13%. However, the use of nanofluids along with the swirl generators increased the Nusselt number by 57.48%. These improvements reduce the thermal losses by 22.3% for swirl generators with nanofluids as opposed to a reduction of only 15.7% with nanofluids alone. The investigation of different swirl generator designs showed various levels of improvements for the overall thermal efficiency and thermal exergy efficiency. The larger swirl generator (H30mm-θ30°-N4) with 6% SiO2 nanofluids was found to be the optimum configuration which improved the overall collector efficiency and thermal exergy by 14.62% and 14.47% respectively. |
1 |
|
(2) In the Introduction part maybe the discussion from page 4, line 81 to page 5, line 131 can be summarized in a Table with advantages and disadvantages. |
The authors appreciate the reviewer’s concerns. This section has been reformulated as requested and Table 1 is now added to the manuscript (part of which is listed below)
…..
|
4-5 |
|
(3) In The Results and Discussion part the following paragraph can be removed: "The numerical results determined in the current study are presented and discussed in this section. [..]This section includes thermal energy enhancement, friction factor, pressure drop, performance evaluation criterion, thermal losses, overall thermal efficiency and energy efficiency." |
This part has been removed. The new presentation becomes:
(4) Results and Discussions
(4.1) Heat Transfer Performance |
22 |
|
(4) From Conclusion part the following text must be removed: "The objective of this study was to improve the 654 thermal and thermodynamic performances of the parabolic trough collector systems". The phrase must be reformulated as a conclusion. Conclusions should be concise and summarize all the important findings of the manuscript. |
This part has been removed from the conclusion section and the conclusions are summarised as shown below Improving the overall efficiency of parabolic trough collectors can lead to enhancement in the sustainable energy extraction and reduction of greenhouse gas emissions. This goal was accomplished using three approaches. The first approach was to investigate the optimum configuration for straight conical strip inserts which were attached to a central rod with varying parameters; pitch height, angle pitch, and the number of strip inserts. The second approach was to introduce the 6% SiO2 nanoparticles mixed with Therminol VP-1 at an inlet temperature of 400K. Finally, the third approach was the combination of the swirl generator insert optimized configuration with nanofluids. The evaluation of these approaches was based on the Nusselt number, pressure drop, performance evaluation criterion, thermal losses, overall thermal efficiency and thermal exergy efficiency. The main findings of this work are summarized below: · The use of straight conical strips enhanced the Nusselt number by 47.13% whereas using nanofluids alone improved the Nusselt number by 15.57%. However, an improvement of 57.48% was observed for Nusselt number by combining the swirl inserts and nanofluids. This combination also resulted in the maximum reduction of the thermal losses by 23.7%. · The improvement in the Nusselt number comes at the expense of an increase in the pressure drop. Swirl generators and nanofluids alone increased the pressure drop by 258.42% and 231.18%, respectively. However, when combined, the pressure drop reached as high as 348.03%. · All the examined cases showed different levels of enhancements in the overall thermal efficiency and thermal exergy efficiency. For the combined case of nanofluid with optimum swirl generator configuration, the overall thermal efficiency improved by 14.62% and the thermal exergy efficiency increased by 14.47%. · The gain in improvement of all tested parameters was found to be more sensitive to the insert geometry and the number of inserts as opposed to the nanofluids. Thus, it can be concluded that swirl inserts are a better candidate than nanofluids for thermal performance improvement in PTC systems. |
34 |
